# The Diagnosis of Sulfated Hemoglobin (SulfHb) Secondary to Sulfur Dioxide Poisoning Using Matrix-Assisted Laser Desorption Time-of-Flight Mass Spectrometry (MALDI-ToF MS)—A Novel Approach to an Unusual Clinical Problem

**DOI:** 10.3390/diagnostics10020094

**Published:** 2020-02-10

**Authors:** Suzanne Docherty, Raminta Zmuidinaite, James Coulson, Martin Besser, Ray Iles

**Affiliations:** 1Department of Haematology, Norfolk & Norwich University Hospital Foundation NHS Trust UK, Norwich NR4 7UY, UK; suzanne.docherty@nnuh.nhs.uk; 2MAP Sciences, Bedford UK, Bedford MK44 3RZ, UK; 3School of Medicine, Cardiff University UK, Cardiff CF10 3AT, UK; CoulsonJM@cardiff.ac.uk; 4Department of Haematology, Cambridge University Hospitals NHS Trust UK, Cambridge CB2 0QQ, UK; martin.besser@addenbrookes.nhs.uk

**Keywords:** sulfated haemoglobin, MetHb SulfHb diagnosis, mass spectrometry, MALDI-TOF MS, oxygen saturation, arterial blood gas analysers

## Abstract

Sulfhemoglobinemia is a rare entity caused by irreversible sulfation of the heme moiety in haemoglobin to form sulfated haemoglobin (SulfHb) and has been caused by H_2_S arising from certain metabolites of drugs and bacterial infection. Clinical presentation is similar to that of methemoglobin (MetHb). Furthermore, it is often difficult to distinguish between the diagnosis of SulfHb from MetHb in arterial blood gas analysers due to the broad overlap in the optical density (OD) absorption spectra—that of SulfHb swamping the more distinct OD absorption shift seen with MetHb. The presence of SulfHb was suspected in a 73-year-old lady with low oxygen saturation (SaO2 ~75%), central cyanosis, and normal arterial oxygen partial pressure (pO2 ~12 kPa). Repeated arterial blood gas analysis on different systems returned error messages for MetHb quantification. There was an improvement in oxygen saturation and cyanosis after an exchange transfusion. A full OD spectrophotometry (500–700 nm) of the patient’s whole blood was suggestive of the presence of SulfHb, with a minor peak absorption at 620 nm. Matrix-assisted laser desorption ionization time-of-flight mass spectrometry (MALDI-ToF MS) was undertaken on whole blood samples from the patient pre- and post-transfusion, alongside normal controls. These demonstrated the presence of SulfHb in the patient’s blood, identifying sulfur, sulfur monoxide, and sulfur dioxide bound to the heme moiety. This gave vital identification as to the cause of Hb sulfation, which was distinct from that previously reported. Levels fell after the exchange transfusion and were completely eradicated after the correct source, an Epsom Salts constipation tonic, was identified. MALDI-ToF mass spectrometry is a new, rapid, specific, and sensitive diagnostic test for rare hematological syndromes such as SulfHb. In addition, it can identify the specific compounds bound to heme. Here, we provide useful diagnostic evidence as to the source of SulfHb, which was via SO_2_ rather than the previously described H_2_S.

## 1. Introduction

A 73-year-old lady presented with a short history of cyanosis following an overseas trip that included two long-haul flights. On examination she was found to have peripheral oxygen saturations of around 75% and cyanosis of her lips, tongue, and fingers, with otherwise normal examination findings. She was initially investigated for possible pulmonary embolism, which was excluded on computerised tomography pulmonary angiography. The patient underwent extensive investigation by the Cardiology and Respiratory services, with an ultimate exclusion of primary cardiac or respiratory pathology as the cause. Her oxygen saturation remained low and was deteriorating on repeated arterial blood gas sampling, despite normal P_a_O_2_ levels (around 12 kPa), and her arterial blood was observed to be chocolate brown in colour (see Figure 1); this did not change when mixed with 100% O_2_ in a syringe. Methemoglobin (MetHb) was suspected [1], but repeated error messages for MetHb quantification were returned by arterial blood gas machines made by different manufacturers in three separate hospitals. 

She was referred to a haematologist for further investigation. Hemoglobin high-performance liquid chromatography (HPLC) and capillary electrophoresis test results were normal. An urgent referral to a National Reference Laboratory for detailed analysis of the patient’s hemoglobin, including sequencing of her globin genes, also returned normal results. Screening for paroxysmal nocturnal haemoglobinuria and glucose-6-phosphate dehydrogenase deficiency were negative. She was prescribed ascorbic acid 200 mg once daily for a working diagnosis of a possible methemoglobin reductase deficiency, which produced no change in her cyanosis. After a subsequent emergency room presentation with fast atrial fibrillation and dyspnoea, the patient underwent an exchange red cell transfusion in the absence of a definitive diagnosis. This resulted in a significant improvement in her symptoms; reduced cyanosis and increased her oxygen saturation but had to be repeated weekly. 

A literature review identified sulfated haemoglobin (SulfHb) as another potential cause of the patient’s clinical signs [2]. She gave a history of rheumatoid arthritis, migraines, and severe constipation, and potential sources of sulfur identified among her medications were the migraine drug rizatriptan, and a compound containing Epsom salts (magnesium sulfate) that was prescribed for constipation. The arterial blood gas machines (manufactured by Roche Global and IL Werfen) available to the authors did not have the capacity to measure SulfHb; the local Clinical Chemistry central laboratory was not able to perform spectrophotometry of haemoglobin. A Clinical Diagnostic Biotechnology start-up company (MAPSciences) had advanced mass spectrometry technology that could rapidly analyse haemoglobin and was approached to undertake analysis of the patient’s blood to establish if SulfHb was detectable. 

## 2. Materials and Methods 

Patient samples were provided as dried blood spots on Guthrie cards and as citrated whole blood. The patient gave written consent for her case to be reported

### 2.1. MALDI-ToF MS of Globins for Aberrant Forms and Adducts

To analyse whole blood samples for globin defects, 1 µL of citrated whole blood was first diluted in 500 µL, 1000 µL and 2000 µL of mass spectral grade distilled deionised water (ddH20) (ROMIL Ltd., Waterbeach, Cambridgeshire, UK). Alternatively, dried whole blood spots were punched with a 4 mm diameter circular hole punch from the center of Guthrie cards. The circular sample was then soaked in 4 mL of ddH20 for 2 hrs and further diluted twice by doubling dilution in ddH20. Matrix-assisted laser desorption ionization time-of-flight mass spectrometry (MALDI-ToF MS) sample slide plates were pre-coated with 1 µl of sinapinic acid (SA) at 10 mg/mL dissolved in 1:2 0.1% trifluoroacetic acid (TFA) and acetonitrile (Sigma Aldrich, Gillingham, Dorset, UK) and were allowed to air dry. One µL of diluted sample was then placed on a well and an additional 1 µL of a matrix was added on a wet sample well. The sample-matrix mixture was left to air dry and form crystals. This sample-matrix mixture was analysed on the Shimadzu 8020 clinical Linear MALDI ToF mass spectrometer that we optimised for analysis at 7000–17000 m/z. MALDI was internally calibrated with a singly and doubly charged Apomyoglobin (Sigma-Aldrich) protein standard [3].

### 2.2. Visible Light Absorption Spectral Analysis of Patient Whole Blood

A total of 20 µL of whole blood samples were lysed in 2 mL of ddH_2_O. Due to the loss of quaternary structure to free subunits and the release of heme moieties from the globin protein chains, not only does this cause red cell lysis but also hemoglobin denaturation. At a dilution of 1/100, the main visible color is due to the abundant heme moieties. Visible spectral absorption was measured at 10 nm intervals between 500 and 700 nm, blanking at each wavelength against pure ddH2O. Spectral absorption was normalized against the peak absorption at 530nm to correct for variation in the total Hb concentration of individual samples and hence permit direct plot comparisons.

### 2.3. MALDI-TOF MS of Blood for Heme Adducts

The control blood samples, M and F, along with the patient’s blood samples at presentation immediately prior to transfusion, post transfusion, and 1 month after magnesium sulfate ingestion was halted were analysed. As described above, 1 µL of whole blood, or dried blood spots soaked in ddH2O, at a dilution of 1 in 2000 was subjected to MALDI-ToF mass spectrometry. MALDI–ToF plates were pre-coated with 1 µL of alpha-cyano-4-hydroxycinnamic acid (CHCA) (Sigma-Aldrich, Bournemouth, Dorset, UK.) at a concentration of 20 mg/mL dissolved in 1:2 0.1% TFA and Acetonitrile, which was allowed to dry to crystals. A total of 1 µL of the diluted patient sample was added, and, before completely drying, another 1 µL of CHCA was added. After the sample matrix mix had completely dried to crystals, this sample was analysed on the Shimadzu 8020 clinical Linear MALDI ToF mass spectrometer, which was optimized for analysis at 500–700 m/z. MALDI was internally calibrated with a singly and doubly charged bradykinin 1–7 fragment (Sigma-Aldrich, ) protein standard.

## 3. Results

The addition of whole blood to ddH_2_O at a dilution of 1 in 2000 not only causes red cell lysis but also the dissociation of hemoglobin into free heme and free alpha and beta globins. Furthermore, as these are the most abundant proteins in lysed blood by more than an order of magnitude, all other molecules are effectively diluted out and these red cell components dominate mass spectral analysis (Figure 2). Sinapinic acid as a matrix preferentially ionizes large proteins and was optimised for globin analysis as previously described [3,4]. The spectral analysis of the patient’s globin at presentation revealed no alteration to the globin proteins. 

As a relatively large amount of citrated blood had been provided, 2 mls of lysed blood was also examined by UV/visible spectroscopy. Two control samples were analysed (one male (control M) and a female (control F) with normal Hb) along with three of the patient’s samples (at presentation, immediately prior to transfusion, and post-transfusion). The expected absorption pattern is two major peaks at 550 nm (OxyHb) and 670–680 nm (DeoxyHb). A comparison of the patient’s admission sample (A) against the control (M and F) spectra shows a large increase in absorption from 600 to 700 nm in the patient’s sample, with a minor maxima/peak at 620 nm (arrowed) (see Figure 3A).

A MALDI-ToF mass spectral analysis of the samples in the CHCA matrix revealed heme moieties at 618 m/z. However, in the patient’s samples prior to transfusion, additional peaks corresponding to heme-sulfur adduct (plus or minus hydrogen, −SH) at 652 m/z, heme-sulfur monoxide (plus or minus hydrogen, −SOH) at 668–670 m/z, and sulfur dioxide (plus or minus hydrogen, SO_2_H) at 683–685 m/z (see Figure 4A). A higher resolution image is shown in Appendix A. These heme-sulfur adduct peaks were greatly reduced in comparison in the control samples and had largely disappeared in the patient’s repeat blood samples post-transfusion (see Figure 4B) and after she had stopped taking magnesium sulfate (see Table 1). 

## 4. Discussion

This case report illustrates the diagnostic difficulties encountered when SulfHb is present and the potential for confusion with MetHb. For this reason, SulfHb has also been known as ‘pseudomethemoglobinemia’ [5]. As demonstrated here, patients typically present with persistently low oxygen saturation on pulse oximetry, normal oxygen partial pressure on arterial blood gas sampling [6,7]. In addition, repeated error messages can be returned by arterial blood gas machines designed to differentiate Methemoglobin. Knowledge of the available arterial blood gas co-oximetry methodology is essential if the presence of SulfHb is suspected, as there is variation in the ability to detect the SulfHb species between co-oximeters produced by different manufacturers—many are unable to differentiate SulfHb and MetHb due to overlap between their absorption spectra [8]. Most notably, SulfHb shares a similar absorption peak with MetHb at 630 nm, and thus a SulfHb may be reported by some systems as a MetHb. Additionally, drugs with both oxidant properties and sulfur groups can produce both SulfHb and MetHb. Gas chromatography has, until now, been considered the ‘gold standard’ technique for the identification of SulfHb [9]. However, as seen in this case, it is frequently not readily available to the investigating clinicians. Other methods employed for establishing the presence of SulfHb are isoelectric focusing and measuring the absorption of light of blood at 630 nm after the addition of cyanide or dithionate (which diminishes absorption by MetHb, but not by SulfHb). The latest generation of co-oximeters are designed to measure SulfHb in addition to MetHb, but these are not, as yet, in widespread use in the authors’ country. The mass spectroscopic technique MALDI-ToF MS was employed after examination visible light absorption spectra of the patient’s whole blood in a commercial lab conclusively demonstrated the presence of SulfHb. Furthermore, this technique successfully identified the presence of sulfur, sulfur monoxide and dioxide bound to heme moieties (rather than hydrogen sulphide), and their reduction following exchange transfusion. The the receipt of a whole blood sample to results took no more than twenty minutes.

SulfHb is a rare complication of exposure of heme groups to sulfur. It causes the irreversible bonding of sulfur to the heme moiety, with resultant cyanosis as SulfHb does not bind oxygen. The clinical presentation is similar to that of MetHb, but it does not respond to treatment with methylene blue or ascorbic acid. However, the presence of SulfHb decreases the oxygen affinity of unaffected hemoglobin, with left-shift of the oxygen-dissociation curve and improved oxygen delivery to tissues, while the converse is true of MetHb; SulfHb thus tends to be associated with milder clinical symptoms than MetHb [10]. There is no specific treatment for SulfHb, and in most cases described to date, clinical symptoms are mild. Where treatment is felt to be required, exchange transfusion is the intervention of choice as it can reduce the proportion of SulfHb given that, once binding of sulfur to hemoglobin has occurred, it will last for the 120-day lifespan of the erythrocyte. This clinical management strategy is reflected in the measured oxygen saturations in this patient (see Figure 5).

From an analytical stand point, true methemoglobin is unlikely to be revealed by this direct MALDI-ToF mass spectrometry approach, as, in this condition, the heme has been oxidized from the normal ferrous (Fe^2+^) to the ferric state (Fe^3+^) and has merely lost an electron, which is a negligible mass difference. 

Gas chromatography mass spectrometry has been reported to differentiate both SulfHb and MetHb from normal heme and this is because separation by this technique is first chromatographic, as a function of column absorption via differences inherent in molecular charge/polarity, and then detection is characterised by mass. However, the technique requires considerable sample pre-preparation (e.g., precipitation of heme from the blood followed by derivatisation), is technically demanding, time consuming, and expensive. Within the clinical setting, optical absorption spectrometry is bedside, rapid, and an inexpensive methodology to determine MetHb. As demonstrated here, MALDI-ToF mass spectrometry is extremely rapid, technically straight forward, negligible in analysis reagent costs, and reveals an adduct to heme that causes a change in mass, such as sulfur and its associated molecular species. 

Interestingly, the “Heme-Sulfo” peaks (labelled in Figure 4) are accompanied by satellite peaks varying in mass by precisely 1 and 2 m/z (labelled with an *). In SulfHb, gaseous hydrogen sulfide or sulfur dioxide bind to the heme of haemoglobin. Although some reports have suggested that the sulfur molecules binds to the porphyrin [10], there is no analytical evidence to support this, and earlier chemists suggest that the interaction is probably via strong ionic, even covalent, linkage to the caged Fe ion [11]. Nevertheless, this results in a significant increase in mass, the difference corresponding to atomic sulfur and oxygen or hydrogen, or both. If chelated to Fe, and not the porphyrin groups of the heme, hydrogen can also bind to the Fe-S complexes and the additional satellite peaks seen in this case may represent hydrogen joining the Heme Fe-S, Heme Fe-SO, and Fe-S0_2_ complexes, i.e., FeSH/FeSH_2_, FeOH/FeSOH_2_, and FeSO_2_H/FeSO_2_H_2_, the further increase in mass being 1 and 2 m/z. (see Appendix A). 

We have also noted that heme, upon MALDI-ToF analysis, is often accompanied by satellite peaks of an increased mass of 1 and 2 m/z presumably via similar interactions. This gives rise to the intriguing possibility that the MALDI-ToF MS technique described may possibly be able to distinguish MetHb by virtue of the fact that the ratio of the 1 and 2 m/z satellite peaks will be different depending on the patient heme being in the normal ferrous (Fe^2+^) or aberrant ferric (Fe^3+^) state. Further studies are ongoing.

For speed and ease of analysis, there is no extraction and a simple lysis, dilute, and shoot was employed. Although haemoglobin is the most abundant molecule in blood, many other components are still seen and one in 1000 or greater dilutions. These are most probably blood group antigens and components of the red blood cell cytoskeleton. Four such minor peaks resolve on the mass spectra close to that of the sulf-heme adducts (labelled with a Ψ) at 658, 672, 689, and 708 m/z, but are currently unidentified. 

## 5. Conclusions

To the authors’ knowledge, this is the first report using MALDI-TOF MS in the detection of SulfHb. Many previous cases of SulfHb described in the literature are thought to relate to hydrogen sulphide causing SulfHb, often in the context of bacterial overgrowth in the gastrointestinal tract (for example, in patients with severe constipation), and by poisoning with agricultural chemicals. MALDI-ToF MS enabled the specific identification of sulfur dioxide rather than hydrogen sulphide bound to heme in this patient’s case, and this assisted with establishing the most likely source of SulfHb in this patient, who both suffers from severe constipation, and was also prescribed drugs containing sulfur groups. MALDI-ToF MS is a highly specific and rapid methodology for the investigation of dyshemoglobin molecules and should be considered as an alternative to gas chromatography for the definitive diagnosis of SulfHb. 

## Figures and Tables

**Figure 1 diagnostics-10-00094-f001:**
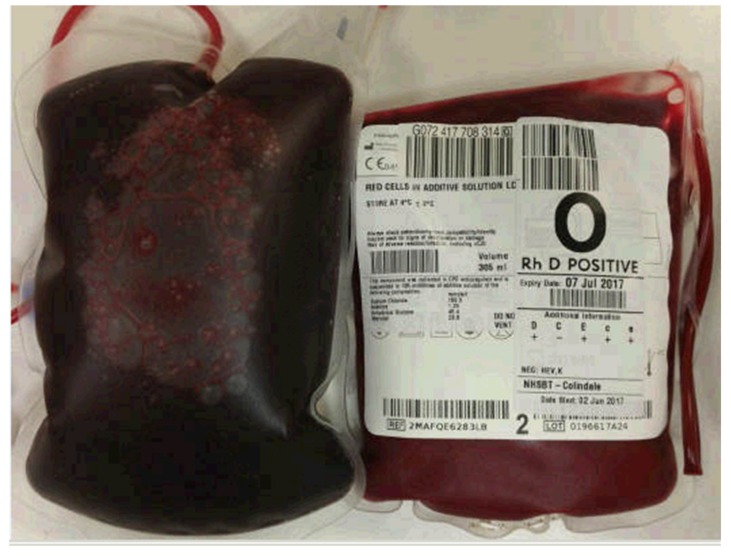
Photograph illustrating the color difference between the patient’s venesected blood (left) and a normal donor unit (right).

**Figure 2 diagnostics-10-00094-f002:**
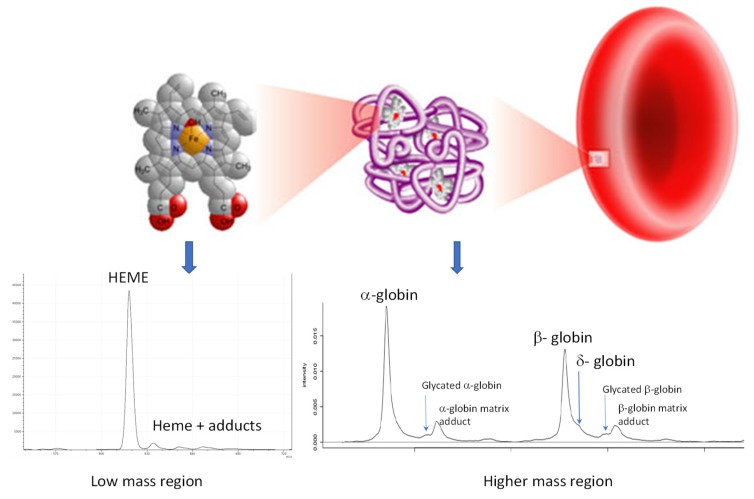
Illustration of lysis of red blood cells by adding whole blood to distilled water, resulting in hemoglobin release, which dissociates into free alpha and beta globin protein chains, along with free heme. These are then resolved simply by matrix-assisted laser desorption ionization time-of-flight (MALDI-ToF) mass spectrometry, thus allowing for the relative quantification of variants and adducts for clinical diagnostic purposes.

**Figure 3 diagnostics-10-00094-f003:**
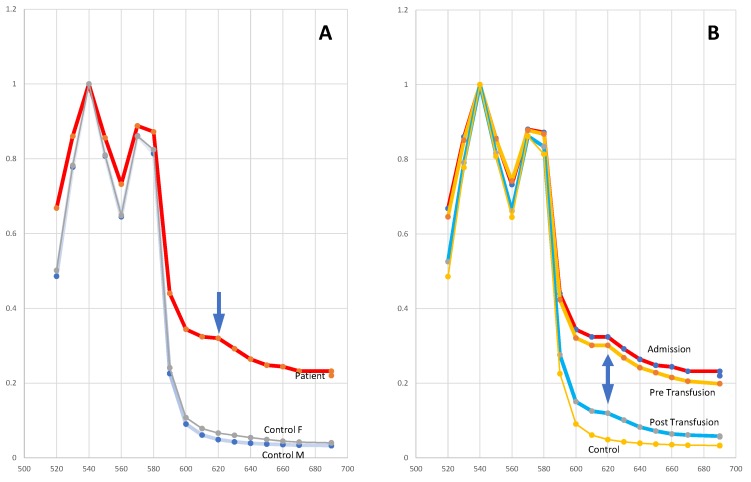
Absorption spectra of hemoglobin samples. Panel (**A**): the patient and a female (F) and male (M) control sample; the arrow marks an increased absorption peak at 620 nm. Panel (**B**): the patient’s sample on admission, pre-transfusion, and post-transfusion compared to the control sample; the double arrow marks the absorption peak at 620 nm seen in all patient samples.

**Figure 4 diagnostics-10-00094-f004:**
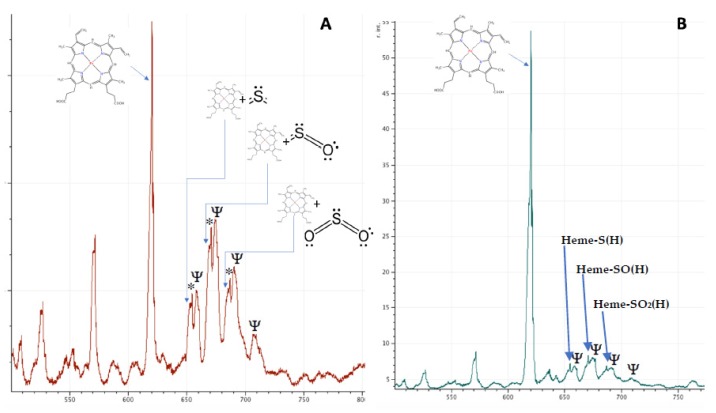
MALDI-ToF mass spectra of patient’s lysed blood demonstrating heme and heme adducts pre- (panel **A**) and post-transfusion (panel **B**). Positions of heme plus sulfur, sulfur monoxide and sulfur dioxide are indicated by the blue arrows. * indicate satellite peaks corresponding to addition of 1 and 2 m/z to the heme-adduct mass. This is probably due to additional hydrogen atoms. **Ψ** indicates peaks corresponding to unknown molecules present in the lysed RBCs but significantly abundant at 1 in 1000 dilution.

**Figure 5 diagnostics-10-00094-f005:**
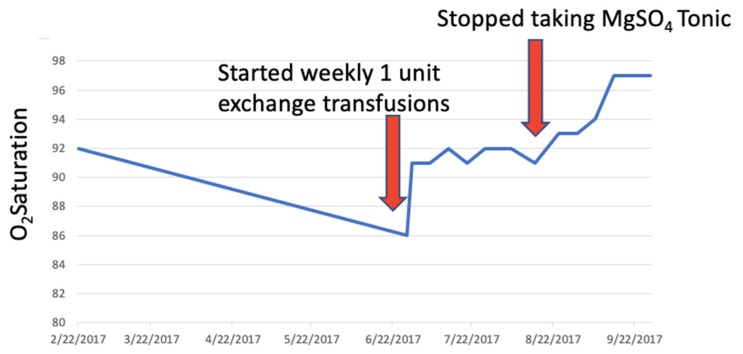
Plot of patient’s oxygen saturation values (blue line) from presentation and during treatment. Red arrows indicate interventions: First exchange transfusion and second cessation of ingestion of magnesium sulfate.

**Table 1 diagnostics-10-00094-t001:** Percentage of heme and sulfated adducts in blood samples.

SAMPLE	Heme (618 m/z)	Heme+ S(H) (652 m/z)	Heme+ S0(H) (669 m/z)	Heme+ S0_2_(H) (684 m/z)
**Male control**	95.3%	2%	2%	0.7%
**Female control**	96.3%	1.4%	1.4%	0.9%
**Patient at presentation**	57.1%	8.8%	18.7%	15.4%
**Patient pre-transfusion**	45.5%	18.2%	21.8%	14.5%
**Patient post-Transfusion**	89.6%	3.8%	3.4%	3.2%
**Patient after cessation of magnesium sulfate treatment**	94%	1.1%	3.5%	1.2%

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
