# Peer review of "The Diagnosis of Sulfated Hemoglobin (SulfHb) Secondary to Sulfur Dioxide Poisoning Using Matrix-Assisted Laser Desorption Time-of-Flight Mass Spectrometry (MALDI-ToF MS)—A Novel Approach to an Unusual Clinical Problem"

_diagnostics, 2020, doi:10.3390/diagnostics10020094_

Round 1
Reviewer 1 Report
Methemoglobin is hemoglobin where the heme has been oxidized from the normal ferrous (Fe2+) to the ferric state (Fe3+) making it incapable of oxygen transport. This leads to a decreased oxygen-carrying capacity and a leftward shift of the oxy-deoxygenated dissociation curve. Oxygen unloading in the tissues is diminished, predisposing to tissue hypoxia. I am not quite sure that this is the first case using MALDI. In 1998, gas chromatography was used (https://jamanetwork.com/journals/jamapediatrics/fullarticle/189784). The methods need to be clarified and be better detailed. Two tables need to be added including one with conditions associated with pseudomethemoglobinemia and one table presenting the pitfalls of gas-chromatography and MALDI.
Author Response
We thanks the reviewer but respectfully point out that this is a case of SulfHb NOT Methemoglobin so commenting tables comparing analysis for MetHb by MS are not appropriate.
In SulfHb gaseous Hydrogen sulfide or sulfur dioxide bind to heme of hemoglobin probably via stong ionic even covalent linkage to the caged Fe ion. This results in a significant increase in mass the difference corresponding to atomic sulphur and oxygen or hydrogen or both (if chelated to Fe) hydrogen can also bind to the Fe-S complex.
Methehemoglobin is an oxidation state whereby the Fe ion becomes 3+ by loss of a further electron. The mass of an electron is far is too small to be detected by MALDI mass spectrometry of a molecule of 618da.
Thus the only comment that can be made in the comparison between a GC-MS analysis and this MALDI MS (for SulfHb) method is that there, no need to extract or derivatise the sample in order to get a result and MALDI is the incredibly fast from sample to result.
Reviewer 2 Report
This manuscript describes the allegedly and as I guess really first investigation of sulphated haemoglobin by MALDI-TOF MS. Hereby the hem group becomes detached from the protein and is detected at around 700 Da. In principle, this could be a straight-forward no-brainer for a reviewer except maybe from a few typos
Unfortunately, the spectra are not isotope resolved and – most regrettably numbers above the peaks are missing. On top of that, the sulfo-peaks are all accompanied by a satellite peak. In the absence of exact mass (no labels on peaks!), speculations on the nature of these doublets cannot be made and to my disappointment the authors did not comment on this phenomenon.
For me the question arises, if this adduct formation could be mimicked in vitro. A very simple experiment that would remove all doubts about the origin of the double peaks.
So, I see this work as interesting but with a few weaknesses that could be ironed out.
The text is rather narrative, especially the introduction. This a bit unusual but also refreshing. No problem.
A few minor things:
L 48: was deteriorating
L 64 and 65: check language
L 73: I would write: the local Clinical Chemistry central lab…
L 88: 1 µl of te matrix
Fig 3: axis labels, increase font size
L 126: White blood cells
Fig. 4: axis labels, increase font size; add masses of peaks, including the doublets
Author Response
Dear Reviewer,
Very happy to correct the typos and also comment on the satellite SulfoHeme-peaks and add the text below in the discussion
"Sulfo peaks (labelled in the Figure) are accompanied by two satellite peaks varying in mass by precisely 1 and 2 m/z. In SulfHb gaseous Hydrogen sulfide or sulfur dioxide bind to heme of hemoglobin probably via stong ionic, even covalent, linkage to the caged Fe ion. This results in a significant increase in mass, the difference corresponding to atomic sulphur and oxygen or hydrogen or both. If chelated to Fe, and not the porphyrin groups of the heme, hydrogen can also bind to the Fe-S complexes and the addition satellite peaks seen in this case may represent hydrogen joining the Heme FeS, Heme Fe-SO and Fe-S02 complexes ie FeSH/FeSH2; FeOH/FeSOH2 and FeSO2H/FeSO2H2; the further increase in mass being 1 and 2 m/z"
Round 2
Reviewer 1 Report
The authors successfully addressed the concerns of the reviewers.
Author Response
Many thanks for your review
Reviewer 2 Report
I still think that the peak apex should get a mass label.
Every peak.
The +1 and +2 Da explanation brought forward in the reply above CANNOT be true. The increment is obviously larger.
It may be that these satellite peaks remain unexplained but they MUST get their m/z.
Unfortunately, the measurements are of poor quality. Unit resolution should be standard nowadays !
Author Response
I fully appreciate the reviewers point, the peaks he is referring to are about 5-10 m/z larger on the spectra. So close but also unknown molecules.
If you resolve the Sulfur heme peaks they will split with "close" satellites +1 and plus 2m/z. I am careful not to imply that the additional hydrogen(s) is charged...it could not be as that would half the m/z in the ToF. Hence I stated dalton but I will change back to m/z.
I will insert a panel into figure 3 showing the 1 and 2+ close satellite peaks and the unknown peaks along with an additional spectra on top of figure 4 of post transfusion blood sample showing dramatically increased heme peak spikes of residual sulfheme and the unknown molecules peaks'.
As we are not purifying, but simply diluting 1/10O0 lysed whole blood, these are likely to be an unrelated abundant molecular species found in RBCs and this will be explained in the revised text.
.....interestingly, looking across the complete mass spectra upto 4000m/z we see significant other molecular mass species that I suspect are the numerous blood group antigens and cytoskeletal components of RBCs. Lot of potential developments!